# Implementing a Diet Risk Score (DRS) for Spanish-Speaking Adults in a Clinical Setting: A Feasibility Study

**DOI:** 10.3390/nu16172992

**Published:** 2024-09-05

**Authors:** Emily A. Johnston, Maria Torres, John Hansen, Kimberly Ochoa, Daniel Mortenson, Elaine De Leon, Jeannette M. Beasley

**Affiliations:** 1Department of Medicine, NYU Grossman School of Medicine, New York, NY 10016, USA; elaine.deleon@nyulangone.org (E.D.L.); jeannette.beasley@nyulangone.org (J.M.B.); 2College of Osteopathic Medicine, California Health Sciences University, Clovis, CA 93611, USA; mtorres@chsu.edu (M.T.); john.hansen@ucsf.edu (J.H.); ochoa2853@chsu.edu (K.O.); mortenson2600@chsu.edu (D.M.); 3Steinhardt School of Culture, Education, and Human Development, New York University, New York, NY 10003, USA

**Keywords:** diet risk score, cardiometabolic disease risk, Spanish language, risk reduction, nutrition in clinical care, diet assessment

## Abstract

Tools to briefly assess diet among US Spanish-speaking adults are needed to identify individuals at risk for cardiometabolic disease (CMD) related to diet. Two registered dietitian nutritionists (RDNs) recruited bilingual medical students to translate the validated Diet Risk Score (DRS) into Spanish (DRS-S). Participants were recruited from a federally qualified health center. Students administered the DRS-S and one 24-h recall (Automated Self-Administered 24-Hour (ASA24^®^) Dietary Assessment Tool) on one day; a second recall was administered within 1 week. Recalls were scored using the Healthy Eating Index (HEI)-2015, a measure of adherence to the Dietary Guidelines for Americans. Spearman correlations, weighted kappa, and ANOVA were conducted using SAS 9.4 to assess the relative validity of the DRS-S. Thirty-one Spanish-speaking adults (female: n = 17, 53%; mean age: 58 (42–69)) completed assessments. The mean DRS-S was 9 (SD = 4.2) (max: 27; higher score = higher risk) and the mean HEI-2015 score was 65.7 (SD = 9.7) (max: 100; higher score = lower risk), with significant agreement between measures (r: −0.45 (*p* = 0.01)), weighted kappa: −0.3 (*p* = 0.03). The DRS-S can be used in resource-constrained settings to assess diet for intervention and referral to RDNs. The DRS-S should be tested in clinical care to assess the impact of dietary changes to reduce CMD risk.

## 1. Introduction

Food is a daily exposure, and dietary patterns can increase or decrease the risk and progression of preventable cardiometabolic disease (CMD) [1]. These conditions significantly increase morbidity, are widespread, and are largely preventable. Prevention [2] and treatment [3,4,5] guidelines recommend behavioral and lifestyle interventions to improve diet, but these are not typically addressed in the time-limited medical office visit [6]. Most healthcare professionals lack the training and the time to conduct a complete diet assessment and counseling, but a brief screening can be completed in most settings without specialized expertise [6].

Hispanic and Latino/a Americans make up the second largest proportion of the US population [7]. Obesity is more prevalent in this population than in the general population in the US. Approximately 42% of the total US adult population has obesity, while 44.8% of Latinos/as and 50.4% of Mexican Americans meet the criteria for obesity (data from NHANES 2017–2018) [8]. Obesity is strongly associated with Obstructive Sleep Apnea (OSA), cardiovascular and cardiometabolic disease, diabetes, and many other health conditions that are in turn affected by modifiable risk factors, such as diet [9]. A validated diet screening tool that is short and easily administered can facilitate early intervention and ease of interpretation of data, determine the need for a more thorough diet assessment, and be a cost-effective approach to disease prevention [10]. Offering a diet screening tool in multiple languages improves accessibility, but few brief diet screeners addressing chronic disease risk have been validated in Spanish [11,12]. The lack of availability of validated Spanish language screening tools exacerbates gaps in health equity and further marginalizes Spanish-speaking patients within the American healthcare system.

The Diet Risk Score (DRS) is a validated nine-item questionnaire created to measure cardiometabolic disease mortality risk related to dietary choices [13]. The DRS scoring is based on comparative risk assessment models and meta-analyses on the impact of foods and food groups on cardiometabolic risk [1,14,15,16,17]. The DRS has been translated into simplified Chinese and validated [18].

The purpose of this study was to assess the relative validity of a brief diet screening tool for Spanish-speaking patients by (1) translating the DRS into Spanish, (2) training medical students to conduct diet assessments, and (3) validating the translated DRS compared to the Healthy Eating Index (HEI-2015) using two 24-h recalls in a group of Spanish-speaking patients receiving care from a Federally Qualified Health Center (FQHC) in California’s Central Valley.

## 2. Materials and Methods

### 2.1. The DRS Questionnaire

The DRS evaluates dietary intake by asking the question: “For the following foods, please select the frequency that best describes how often you eat each food or group of foods in a normal week”. Participants can select from the following options: “daily”, “2–3 times per week”, “1 time per week”, and “never”. Every response is given a rating from 0 (minimum) to 3 (maximum), resulting in a total risk score of 0–27. The details of the creation and validation of the DRS have been published [13]. Two bilingual medical students translated the DRS into Spanish. The translated document was then reviewed and approved by a Medical Spanish faculty member and a bilingual Spanish-speaking registered dietitian nutritionist (RDN). Corrections were made based on the feedback, and the questionnaire was finalized.

### 2.2. Study Approval and Informed Consent

This study was approved by the California Health Sciences University (CHSU) Institutional Review Board. All potential participants completed written informed consent prior to beginning any study activities.

### 2.3. Recruitment 

Spanish-speaking adults between the ages of 35 and 75 years old were recruited from a federally qualified health center (FQHC) between October 2022 and February 2023. This age group has a higher risk of CMD than younger adults [19], and is consistent with the prior DRS studies [13,18]. An RDN and a health educator at the FQHC called patients who met the eligibility criteria to explain the study to them and ask if they were interested in participating. If they agreed, they were scheduled for an in-person visit at the FQHC. Participants received a USD 25 Walmart gift card to acknowledge their participation in the study. Gift cards were donated by the study site administration.

### 2.4. Training 

An RDN trained medical students to conduct 24-h recalls and to administer the translated DRS (DRS-S). This included teaching the medical students how to ask detailed questions about ingredients, cooking methods, and portion sizes to ensure accurate data collection. Students were native and non-native bilingual Spanish speakers. Additionally, the training emphasized the importance of creating a comfortable and non-judgmental environment for participants to openly share their dietary habits. The students read, translated, and used an abbreviated version of the USDA Automated Multiple-Pass Method to improve the accuracy of data collection [20]. The students practiced using the Automated Self-Administered 24-Hour (ASA24^®^) Dietary Assessment Tool, version (2020), developed by the National Cancer Institute, Bethesda, MD [21], to assess their own diets. They also practiced using the DRS-S on themselves and assessed each other’s diets via virtual Zoom meetings while the RDN observed and provided feedback.

### 2.5. Data Collection

At the scheduled appointment, the student or health educator reviewed the informed consent and study details with the participants. If the participant consented, a medical student conducted an in-person administration of the initial 24-h dietary recall using the ASA-24 and the DRS-S for each participant at the FQHC with an RDN present. Portion size tools were available for patients to visualize the food portions consumed [22]. An RDN or trained bilingual health educator conducted the second 24-h dietary recall with the participant by telephone. Medical students entered 24-h dietary recall data into the ASA24 website. 

### 2.6. Statistical Analysis

We scored the DRS-S according to the original DRS scoring from 0 (low risk) to 3 (high risk) for each component, and calculated a total score out of 27 for each participant (Appendix A). We applied the HEI scoring macro from the National Cancer Institute [23] to score the 24-h recalls using the Healthy Eating Index (HEI)-2015, which measures adherence to the Dietary Guidelines for Americans [24]. The HEI macro provides scoring by HEI component (e.g., Total Fruits, Whole Fruits) and a total score from 0 (zero adherence to the 2015–2020 Dietary Guidelines for Americans; high risk) to 100 (perfect adherence to the 2015–2020 Dietary Guidelines for Americans; low risk). We used SAS 9.4 (SAS Institute, Cary, NC, USA) for data analysis and performed Spearman correlations (PROC CORR) between total and component scores. We also evaluated strength of agreement between tertile rankings through weighted kappa and used one-way analysis of variance (ANOVA-PROC ANOVA) to determine statistical differences between DRS-S and HEI-2015 total score tertiles. 

## 3. Results

Thirty-two Spanish-speaking adults completed this study. One participant completed only one 24-h recall and was excluded from the analyses. All questionnaires were administered in Spanish. No participants were excluded for over or underreporting energy intake. The mean age of the participants was 58 (range 42–69) years, and 53% were female (Table 1). The mean DRS-S score of respondents was 9 (SD = 4.2) out of a maximum score of 27. The mean HEI-2015 score was 65.7 (SD = 9.7) out of a maximum of 100 (Table 1). 

Mean scores for adequacy components out of 10 were 8.1/10 for fatty acids (mono and polyunsaturated fats), 2.8/10 for whole grains, and 3.5/10 for dairy. Mean scores for moderation components of the HEI-2015 were 4.3/10 for sodium intake, 4.1/10 for refined grain intake, 7.8/10 for saturated fat intake, and 9.5/10 for added sugar intake (Table 1). Higher scores reflect a more desirable intake for all HEI components. Figure 1 demonstrates the degree of alignment with recommendations for each of the components of the HEI-2015 score stratified by total HEI-2015 score tertile.

The DRS-S and HEI-2015 total scores were inversely correlated (r = −0.44, *p* = 0.01). DRS-S fruit and HEI-2015 total fruit component scores were also inversely correlated (r = −0.45, *p* = 0.01) (Table 2). Participants in the highest tertile of DRS-S score had a lower HEI-2015 score compared to those in the lowest tertile of DRS-S score (HEI-2015 = 58.2 versus 70.8, *p* = 0.01) (Table 3). 

## 4. Discussion

We translated a brief diet screening tool into Spanish and tested it in a group of Spanish-speaking patients who receive care at an FQHC in Central California. Medical students assessed diets and collected data after they were trained in and practiced assessing diets. The DRS-S was inversely correlated with the HEI-2015 score and the two measures stratified risk similarly. The total fruit component scores from the DRS-S and HEI-2015 were also inversely correlated. We found that participants’ reported dietary intake reflected overall poor adherence to the Dietary Guidelines for Americans. The mean HEI-2015 score was 65.7 out of 100, slightly above the population average of 58 out of 100 [25]. The mean DRS-S score was 9 out of 27, which represents a low-moderate CMD risk related to diet. 

Assessing diet among Hispanic and Latino/a individuals can be challenging as this demographic represents a heterogeneous group from Mexico, Central America, South America, and the Caribbean. Diet quality scores for people of Mexican heritage and also those of Dominican and Central American heritage may be higher compared to other Hispanic/Latinx groups [26]. The Hispanic Community Health Study/Study of Latinos (HCHS/SOL) found that the average HEI-2010 score among Mexicans was 71.1 (SD = 0.86) [26], higher than our study population average of 65.4 (SD = 9.7). In a study comparing diet quality among Hispanic men and women, Hispanic men had lower overall diet quality and component diet quality scores based on intake of healthy food groups [27]. 

Few brief, validated, Spanish-language diet screeners are available [28]. The USDA Food Behavior Checklist was translated into a 22-item tool and validated among Spanish-speaking adults in California [28,29] with the goal of assessing the effectiveness of USDA community nutrition interventions. There are several Block screeners available in Spanish that query fats, fruits/vegetables/fiber [30], and folic acid intake, all via separate questionnaires. The 13-item Latino Dietary Behaviors Questionnaire (LDBQ) was validated against 24-h recalls [11] and can be administered and scored relatively quickly; it includes questions about behaviors (e.g., eating a complete breakfast) and foods (e.g., drinking 1% or skim milk) and does not include others, such as intake of fish, nuts, and processed meats, that may be predictive of diet risk. The 14-item Mediterranean Diet Adherence Screener (MEDAS) was validated in older Spanish adults in Spain [31] and other European countries [32], but has been used minimally in the US in Spanish-speaking adults. A 15-item questionnaire can be considered brief compared to a full food frequency questionnaire, but could take 10 min to complete, which may be the length of a follow-up clinical visit [12]. The DRS-S is brief: it was completed in approximately 2 min on average in the English validation study [13] and in approximately 1 min in a medical student demonstration. The DRS-S is easy to score and provides talking points for clinicians to make brief recommendations for change based on the riskiest reported dietary components. It is based on current evidence on diet and CMD risk, which can be of use in the time-limited clinical setting.

Physicians have an opportunity to engage in health promotion in clinical care [33]. However, nutrition and diet are insufficiently incorporated into medical education in medical schools around the world [34,35,36]. In today’s complex and time-limited healthcare setting, interactions between doctors and patients are limited. Outpatient office visits have a median visit length of 15.7 min, covering a median of six topics [37], where each topic may receive just 1 min of the visit. Due to increasing patient loads, doctors are trained to emphasize one or a few chief complaints. Nutrition counseling is rarely provided, and time spent on it is inadequate to initiate behavior change [35,38,39]. 

Healthcare providers from different disciplines can work together on health promotion for the betterment of patients. If diet quality is poor, asking simple but specific questions can help to identify challenges to health promotion that can be discussed and intervened upon in the healthcare setting. In our sample, 48% reported consuming processed meats and 62% reported consuming sugar-sweetened beverages at least once per week, while 22.6% of participants reported never eating nuts and 19.4% reported never eating fish. However, 74.2% reported consuming vegetables every day and 64.5% reported consuming fruits every day (Appendix B). Rather than providing generic dietary advice, e.g., “eat more fruits and vegetables”, recognizing that individualizing suggestions to the pressing issues of the patient can save time and be more effective, e.g., “reduce intake of sugar-sweetened beverages and processed meats”.

This study was conducted in the Central Valley of California, which provides food for the entire nation, but is also rife with food deserts and has high rates of hunger [40]. Osteopathic medical students were involved in translating the DRS into the DRS-S, data collection, and data entry. This is important as many osteopathic physicians become primary care providers [41], often a patient’s first encounter with the healthcare system, but rates of nutrition education [42] and knowledge [43] among osteopathic medical students are low. Equipping future physicians with methods to quickly and accurately assess diet in the outpatient setting will help improve patient care. This study was performed at an FQHC, where an RDN was available at primary care clinic visits, which is not the case in all medical practices. A recent study surveying family practice physicians revealed that over half of respondents did not have an RDN on-site (64%) yet were highly interested in integrating an RDN (94.9%) [44]. Even in a resource-limited healthcare setting, physicians can start the discussion of nutrition, guided by a tool like the DRS-S, and refer patients with poor scores to RDNs who can provide ongoing nutrition counseling and support.

This project integrates three previously uncoupled approaches to diet-related disease risk: the creation of a brief but effective and validated diet-risk assessment, sensitivity to language, and medical student training to apply this tool in a primary care setting. The ASA-24 questions were administered by students and health educators, not self-administered, due to the researchers’ experience with participant confusion related to self-administration. We used a well-accepted diet assessment tool as the comparator for this validation study. Medical students noted that during interviews, participants frequently reported eating *carne asada*, roasted or grilled steak, but few were able to estimate the serving size. Additionally, they noted that while *enchiladas verdes, rojas, potosinas, norteñas, Michoacanas*, and *suizas* are different versions of the same food, home preparation methods may vary widely, resulting in large differences in calorie and nutrient consumption, and students often had to enter recipes manually if they could not find the foods in the database. Students reported the importance of being familiar with foods commonly consumed by patients as they otherwise would not have known to ask follow-up questions. For example, participants frequently reported eating *caldo de camarón* (shrimp soup), which is commonly accompanied by tortillas. By understanding the cultural significance of tortillas in Latin cuisine, students were able to specifically inquire whether meals were consumed with or without tortillas, and if they were included, they further asked about the quantity and type of tortillas consumed. 

This study has several limitations. Of note, some data collection took place during Lent, a time when followers of religious rules may abstain from consuming meat on Fridays and restrict other foods. It is possible that the Lenten dietary restrictions may have had a positive, negative, or negligible influence on the reported dietary intake. We did not collect demographic data beyond age and sex, nor did we collect medical history, data on CMD risk, or other related factors. This study had a small sample size; however, power calculations from prior work show that this sample was adequately powered for analyses [18]. While our study obtained data from a total of 31 participants, this may not be representative of individuals outside of the Central Valley of California. Due to limited participants, subgroup analyses could not be performed.

## 5. Conclusions

While further work is required to test and implement the DRS-S, this is an important step in creating more inclusive tools to promote discussion of modifiable risk factors in clinical care. Training clinicians in brief assessment and counseling as well as providing tools to guide the discussion, can lead to reductions in preventable disease risk related to diet. Providing language-congruent tools can help healthcare providers initiate discussions with non-English speaking patients on ways to reduce their risk of preventable chronic diseases. 

## Figures and Tables

**Figure 1 nutrients-16-02992-f001:**
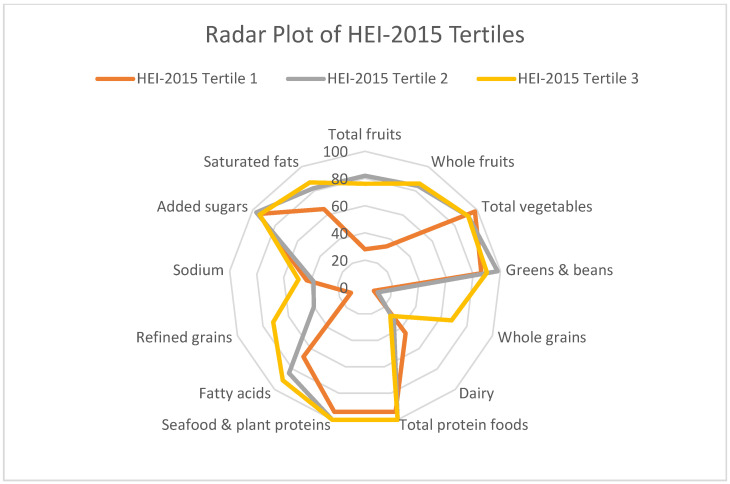
Radar plot of HEI-2015 scores by tertiles. Outer edges of circle reflect greater adherence to each component score.

**Table 1 nutrients-16-02992-t001:** Descriptive characteristics, n = 31.

Variable	Mean	95% CI
Age (±SD)	58 years (±8.4)	
Sex	52% female (n = 16)	
Reported total energy (ASA-24; calories)	3282	2965, 3600
DRS-S Fast food	0.8	0.6, 1.1
DRS-S Bread	1.6	1.2, 2.0
DRS-S Snacks	0.9	0.6, 1.2
DRS-S Processed meat	0.9	0.4, 1.4
DRS-S Sugar-sweetened beverages	1	0.6, 1.4
DRS-S Nuts	0.8	0.3, 1.3
DRS-S Fish	1.1	0.7, 1.5
DRS-S Veg	0.8	0.3, 1.3
DRS-S Fruit	1.2	0.6, 1.7
DRS-S Total Score	9	7.4, 10.5
HEI-2015 Total Vegetables *	4.7	4.4, 5
HEI-2015 Greens/beans *	4.6	4.1, 5
HEI-2015 Total fruit *	3.1	2.3, 3.9
HEI-2015 Whole fruit *	3.4	2.7, 4.2
HEI-2015 Whole grains **	2.8	1.5, 4.1
HEI-2015 Dairy **	3.5	2.4, 4.6
HEI-2015 Total protein *	4.9	4.8, 5
HEI-2015 Seafood/plant proteins *	4.9	4.7, 5
HEI-2015 Fatty acids **	8.1	7.2, 9
HEI-2015 Sodium **	4.3	3.2, 5.5
HEI-2015 Refined grains **	4.1	2.6, 5.5
HEI-2015 Fatty acids (SFA) **	7.8	7, 8.6
HEI-2015 Added sugars **	9.5	9.2, 9.8
HEI-2015 Total Score	65.7	62.2, 69.3

ASA-24: Automated Self-Administered 24-Hour Dietary Assessment Tool; DRS-S: Diet Risk Score-Spanish; HEI: Healthy Eating Index; SFA: saturated fatty acids. * HEI range: 0–5; ** HEI range: 0–10.

**Table 2 nutrients-16-02992-t002:** Correlation between DRS-S and HEI-2015 component scores.

DRS-S Component	HEI-2015 Component	Correlation	*p* Value
Fast food	Sodium	0.20	0.27
Bread		−0.27	0.13
Snacks		0.15	0.41
Processed meats		−0.10	0.56
	Saturated fat	0.09	0.59
Sugar-sweetened beverages	Added sugars	−0.03	0.86
Nuts	Seafood/plant protein	−0.24	0.18
Fish		−0.008	0.96
**Fruit**	**Total fruit**	**−0.45**	**0.01**
	Whole fruits	−0.41	0.02
Vegetables	Total vegetables	0.06	0.72
	Green vegetables, beans	0.01	0.93
**Total DRS-S**	**Total HEI-2015 score**	**−0.44**	**0.01**

DRS-S: Diet Risk Score-Spanish; HEI: Healthy Eating Index. Bold components are statistically significant.

**Table 3 nutrients-16-02992-t003:** Mean HEI-2015 score by DRS-S category.

HEI Scores by DRS-S Tertile	
DRS-S Tertile	HEI Scores (95% CI)
1 (0–9)	70.8 (65.5, 76.1) *
2 (10–18)	66.1 (61.0, 71.2)
3 (19–27)	58.2 (52.0, 64.5) *

DRS-S: Diet Risk Score-Spanish; HEI: Healthy Eating Index. * Data presented as means (95% CIs) from one-way ANOVA; * tertiles 1 and 3 are significantly different (*p* = 0.01). ANOVA: analysis of variance.

## Data Availability

Data are available upon reasonable request to the corresponding author due to time limitations.

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
