# Peer review of "Implementing a Diet Risk Score (DRS) for Spanish-Speaking Adults in a Clinical Setting: A Feasibility Study"

_nutrients, 2024, doi:10.3390/nu16172992_

Round 1
Reviewer 1 Report
Comments and Suggestions for Authors
Dear Authors,
The manuscript "Relative Validity of a Diet Risk Score (DRS) for Spanish 2 Speaking Adults", seems to be interesting and well presented. The below mentioned points need to be resolved.
- Line 2: It is better to add the study design in the title to give the readers a brief overview of the study. Add dash (-) between ‘Spanish-speaking’
- Abstract: The main problem and urgency, as well as the results, are well described.
- Line 23: ‘This must be tested in clinical care.’ This sentence could be rephrased to sound softer.
- Line 37: It would be better if the authors stated numeric prevalence data.
- Line 53-58: The purpose of the study is well-described.
- Line 68-70: ‘This study was approved by the California Health Sciences University (CHSU) Institutional Review Board. All participants completed informed consent.’ — Relocate to a new sub-section named ‘Study Approval and Participants’ Informed Consent’.
- Line 70-72: Reward for participants should be relocated to the ‘Recruitment’ sub-section.
- Line 88: It is better to add ‘video call meeting’ after ‘Zoom’ because not all readers would understand.
- Line 96: Just write ‘ASA24’ because the abbreviation is already written above.
- Line 108: Change into ‘(ANOVA—PROC ANOVA)’
- Line 129-131: Better to add the abbreviations for Table 1 & Table 2.
- Line 149: Delete the excessive tab before the paragraph.
- Better to add ‘=‘ between SD and SD value throughout the manuscript.
- Suggestions for dietary education are well-described in the discussion.
- Line 223: ‘caldo de camarón’ make itallic.
- Line 227: Before ‘Of note’, enter to make a new paragraph specifically explaining the study's limitations.
- Line 240: change the second ‘and’ before ‘providing tools’ into ‘as well as’ to improve the readability.
Comments on the Quality of English Language
minor errors need to be addressed
Author Response
Dear Reviewers,
Thank you for your thoughtful comments on our submission, "Relative Validity of a Diet Risk Score (DRS) for Spanish Speaking Adults". Our responses to your critiques are below in Italics and noted in the manuscript in tracked changes.
Reviewer 1
- Line 2: It is better to add the study design in the title to give the readers a brief overview of the study. Thank you for this comment. We changed the title to: Implementing a diet risk score for Spanish-speaking adults in a clinical setting: a feasibility study.
Add dash (-) between ‘Spanish-speaking’ Done, thank you.
- Abstract: The main problem and urgency, as well as the results, are well described. Thank you.
- Line 23: ‘This must be tested in clinical care.’ This sentence could be rephrased to sound softer. Thank you, we changed the last sentence of the abstract to: The DRS-S should be tested in clinical care to assess impact on dietary changes to reduce CMD risk.
- Line 37: It would be better if the authors stated numeric prevalence data. We added percentages to this statement.
- Line 53-58: The purpose of the study is well-described. Thank you.
- Line 68-70: ‘This study was approved by the California Health Sciences University (CHSU) Institutional Review Board. All participants completed informed consent.’ — Relocate to a new sub-section named ‘Study Approval and Participants’ Informed Consent’. Done.
- Line 70-72: Reward for participants should be relocated to the ‘Recruitment’ sub-section. Done.
- Line 88: It is better to add ‘video call meeting’ after ‘Zoom’ because not all readers would understand. We clarified this, thank you.
- Line 96: Just write ‘ASA24’ because the abbreviation is already written above. Done, thank you.
- Line 108: Change into ‘(ANOVA—PROC ANOVA)’ Done.
- Line 129-131: Better to add the abbreviations for Table 1 & Table 2. Done.
- Line 149: Delete the excessive tab before the paragraph. Done.
- Better to add ‘=‘ between SD and SD value throughout the manuscript. Done, thank you.
- Suggestions for dietary education are well-described in the discussion. Thank you.
- Line 223: ‘caldo de camarón’ make itallic. Done.
- Line 227: Before ‘Of note’, enter to make a new paragraph specifically explaining the study's limitations. Done.
- Line 240: change the second ‘and’ before ‘providing tools’ into ‘as well as’ to improve the readability. We edited this sentence, thank you.
Reviewer 2 Report
Comments and Suggestions for Authors
Dear Authors,
Your article is of interest for American Spanish speaking population better healthcare service, but we have our main concerns with the title. For us this is not a validation for Spanish speaking adults, first because the population used is not representative of Spanish speaking adults, not only for the reasons you invoked about the diversity of Spanish speaking population in US, but mainly because the population used is a group with higher risk of CMD, recruited in a FQHC, and that abstained a score of 9 that represent moderate risk, probably because they already received some medical care or whatever the reason are very far of the higher nutritional risk, what the questionnaire aim to detect. Another reason is that you used students not sufficiently accustomed to Spanish speaking dietary habits population for the 24 h recalls. We understand why you used those students, to prove that not specialist can use the questionnaire and obtain useful information’s in very short time, but for the 24 h recalls it is a big risk for the validation of the Spanish version of the questionnaire. Those observations do not invalidate your study, but the title must be reviewed to be the reflect of what you effectively did: “evaluate the validity or feasibility to use students not specialised in the use of a Spanish translation of the DRS in CMD Spanish speaking population, as a pilot study for future validation do the DRSs as an easy and rapid use tool …”, or something like that.
Lines 38 to 40, please put a reference for those affirmations.
Lines 46 to 47 we suggest “within the American healthcare system”.
Material and methods, HEI procedure is not explained, please correct this with references and normative values, highlighting that the higher the score, the lower the risks in contrast to the DRS.
Lines 66 to 68, please put reference for this procedure.
Line 81, please put reference for 24 h recall procedure.
Lines 85 to 88, please put reference for those procedures.
Line 93, please put reference for portion size tool used in the study.
Lines 153 to 157, you should specify that those studies were in general population, while your study were in higher CMD risk population.
Lines 178 to 236, this text is more linked to the limits of the study and should be reduced to transmit the same ideas in a common title like: “limits of the study and suggestion for future studies”. Probably a limit for the validation of the questionnaire should be the fact to use the students for the 24 h recall and maybe you should explain why you did use this option.
Lines 216, carne asada is not grilled it is roast
In the abstract you finish by “ This must be tested in clinical care.” Please add some reason to be sure that future lecturer will not interpret that the DRSs is completely validate by the present study.
References 12 and 30 are for the same article, please check if they are other duplicates.
Wishes of success.
Author Response
Dear Reviewers,
Thank you for your thoughtful comments on our submission, "Relative Validity of a Diet Risk Score (DRS) for Spanish Speaking Adults". Our responses to your critiques are below in Italics and noted in the manuscript in tracked changes.
Thank you for your feedback. We changed the title to: Implementing a diet risk score for Spanish-speaking adults in a clinical setting: a feasibility study.
Eligible participants were adults over the age of 35 who were patients of the FQHC and spoke Spanish, therefore we did not characterize our participants as high-risk. We appreciate your comments that these criteria still limit our ability to validate the DRS for all Spanish speaking adults and make this clear in the discussion.
Our team included students, Registered Dietitians, and a health educator who are bilingual and familiar with the food and culture of the participants. Students were trained by the Registered Dietitians and observed by them to ensure accurate data collection, although bias does exist in all self-reported data, which we highlight in the limitations.
Lines 38 to 40, please put a reference for those affirmations. Done.
Lines 46 to 47 we suggest “within the American healthcare system”. Done.
Material and methods, HEI procedure is not explained, please correct this with references and normative values, highlighting that the higher the score, the lower the risks in contrast to the DRS. We have added more detail to this section.
Lines 66 to 68, please put reference for this procedure. We added an additional sentence to help clarify the process. There is no specific reference, it occurred as described in the methods.
Line 81, please put reference for 24 h recall procedure. Done.
Lines 85 to 88, please put reference for those procedures. Done.
Line 93, please put reference for portion size tool used in the study. Done.
Lines 153 to 157, you should specify that those studies were in general population, while your study were in higher CMD risk population. Please see the response above.
Lines 178 to 236, this text is more linked to the limits of the study and should be reduced to transmit the same ideas in a common title like: “limits of the study and suggestion for future studies”. Probably a limit for the validation of the questionnaire should be the fact to use the students for the 24 h recall and maybe you should explain why you did use this option. We added to a line to the discussion about our choice to have students administer the ASA-24.
Lines 216, carne asada is not grilled it is roast. Thank you for this comment. Since carne asada can be prepared by either grilling or roasting the meat, we have made this correction in the manuscript.
In the abstract you finish by “This must be tested in clinical care.” Please add some reason to be sure that future lecturer will not interpret that the DRSs is completely validate by the present study. Thank you, we changed the last sentence of the abstract to: The DRS-S should be tested in clinical care to assess impact on dietary changes to reduce CMD risk.
References 12 and 30 are for the same article, please check if they are other duplicates. Thank you, we have edited this and checked all references for duplicates.
Wishes of success. Thank you!
Round 2
Reviewer 2 Report
Comments and Suggestions for Authors
Dear Authors,
thank you for taking in count all our recommendations’, about assada, in Portuguese it is not grilled, it is only roasted, reason for my confusion for Spanish.
Best regards